# Relative contributions of the correlates of stunting in explaining the mean length-for-age z-score difference between 24-month-old stunted and non-stunted children living in a slum of Dhaka, Bangladesh: results from a decomposition analysis

Subhasish Das,[1] Md Ashraful Alam,[1] Mustafa Mahfuz,[1] Shams El Arifeen,[2] Tahmeed Ahmed[1]

For numbered affiliations see end of article.

**Correspondence to**
Dr Subhasish Das;
subhasish.das@icddrb.org

## ABSTRACT

**Objective** Using MAL-ED (Etiology, Risk Factors, and Interactions of Enteric Infections and Malnutrition and the Consequences for Child Health) Bangladesh birth cohort data, we sought to measure the relative contributions of the most predictive correlates of stunting to mean length-for-age z (LAZ) score difference between stunted and non-stunted children at 24 months of age.

**Setting** Dhaka, Bangladesh

**Participants** 211 slum-dwelling children enrolled within 17 days of their birth.

**Variables and method** The explanatory variables were identified from the following groups: maternal characteristics, birth characteristics, macronutrient intake, socioeconomic status, morbidity and serum micronutrient level. At step 1, predictive correlates of stunting were identified longitudinally (from 9 to 24 months of age) using generalized estimating equations (GEE) model. Then, the relative contributions of the most predictive correlates of stunting to mean LAZ score difference between stunted and non-stunted children at 24 months of age was measured using Blinder-Oaxaca decomposition analysis

**Results** The GEE multivariable model identified maternal height, birth weight, people per room, gender, having separate room for kitchen and energy intake as the most predictive correlates of stunting. At 24 months, mean LAZ score difference between stunted and non-stunted children was 1.48. The variable by variable decomposition of the LAZ gap identified maternal height (coefficient: −3.04; 95% CI: 0.35 to -6.44), birth weight (coefficient: −0.21; 95% CI: 0.88 to -1.30), people per room (coefficient: 0.31; 95% CI: 0.92 to -0.30) and energy intake (coefficient: −0.12; 95% CI: 0.22 to -0.46) as the top most factors responsible for the mean LAZ score difference between stunted and non-stunted children at 24 months of age.

### Strengths and limitations of this study

► This study used the threefold Blinder-Oaxaca decomposition analysis to decompose the relative contributions of most predictive correlates of stunting to mean LAZ score difference of stunted and non-stunted children into three components: endowments, coefficients, and the interaction of the two.

► The explanatory variables used for the decomposition analysis were identified after analysing the longitudinal birth cohort data collected from the same children.

► At 24 months, 20% of the enrolled children were lost to follow-up which could lead to possible bias.

► The findings of this study might not be applicable to the non-slum setting, as this study was conducted in a slum area of Dhaka, Bangladesh.

**Conclusions** The relative contributions of maternal height and birth weight to LAZ gap signifies that improvement in nutritional status of a women during her adolescence and pregnancy would have an impact on birth weight of her offspring, and ultimately, on linear growth of the child.

## INTRODUCTION

Stunting (length-for-age more than 2 SD below the WHO Child Growth Standards median) is an indicator of chronic under-nutrition and is one of the best measures of child health inequality.[1] It is one of the most prevalent forms of malnutrition which is known to be associated with higher mortality and cognitive impairment of affected children.[1–4] Worldwide, 150.8 million under-five

children are stunted and in Asia, the prevalence is 23.2%.[5] Variation in rates of stunting exists not only across continents but also within local geographies.[6] The prevalence of stunting in Bangladeshi slum areas is much higher than the national level average (50% vs 36%, respectively).[7 8]

Children from lowest wealth quintiles who have a normal anthropometric status at birth may experience an intense period of growth retardation.[9] During the subsequent life-cycle stages, this growth retardation continues to exert negative effects such as socioemotional impairment in preschool age, poor school attainment in early childhood and higher risk of chronic disease in adulthood.[10] WHO conceptual framework on childhood stunting reported maternal factors, home environment, inadequate complementary feeding, lack of breast feeding and infection as the most proximal factors responsible for stunting.[11] But, relative contributions of these correlates to length-for-age z (LAZ) score difference between stunted and non-stunted children were not measured. In high-burden countries, the benefit–cost ratio of investments to reduce stunting ranges from 3.8 to 47.9, whereas the ratio is 18.4 in Bangladesh.[10] To that end, since Bangladesh is a resource-constrained country, identifying the top contributors of stunting would help the policy-makers to direct concerted stunting prevention programme in a cost-effective way. Keeping the context in mind, we identified the correlates of stunting among children aged 9–24 months using the longitudinal data emanated from a birth cohort study. Then, we measured the relative contributions of the most predictive correlates of stunting in explaining the mean LAZ score difference between stunted and non-stunted children at 24 months of age.

## METHODS
### Study participants
Data used for this analysis were collected from the MAL-ED (Etiology, Risk Factors, and Interactions of Enteric Infections and Malnutrition and the Consequences for Child Health) Bangladesh site. The MAL-ED birth cohort study was conducted at eight sites in three continents. In Bangladesh, the MAL-ED study site was Bauniabadh slum, Mirpur, Dhaka, which was described elsewhere.[12] Following a well-defined protocol, field workers visited households and enrolled 265 healthy newborn children within 17 days of their birth between February 2010 and February 2012. Participants were followed until 24 months of age. At 24 months, 211 participants remained which is 80% of the total recruitment.

### Data collection
#### Dependent variable
Following standard anthropometric methodology, trained field workers measured the recumbent length of the children to the nearest 1 mm using Seca infantometer (model no: 417, Hamburg, Germany) every month until 24 months of age.[13] The LAZ score of each child was determined using the WHO 2006 child growth standards.[14] A child with LAZ score below −2 is classified as stunted.[14]

### Independent variables
The explanatory variables used for the analysis were drawn from the following groups: birth characteristics, maternal characteristics, nutrient intake, socioeconomic status, morbidity and serum micronutrient. Data on birth characteristics (enrolment weight and length) and maternal height were collected within 72 hours of recruitment. Children, wearing minimum clothes, were weighed using a digital scale with 10 g precision (Seca, model no: 345). The enrolment weight was used as the surrogate measure for birth weight. Standard wooden height measuring boards were used for measuring maternal height. For assessing the feeding habit and energy intake of the children, 24-hour food frequency data were collected monthly from ninth month onwards using a 24-hour multiple-pass dietary recall approach.[15] The 24-hour dietary recall interviews were conducted on non-consecutive days and out of every four recalls, one interview was conducted on a weekend. A locally adapted food composition table was used to convert the dietary intake data to energy.[16] Minimum acceptable diet (MAD)—a core indicator of infant and young child feeding practice was used to measure the appropriateness of the complementary feeding practice of the children.[17] MAD meets standards for both minimum dietary diversity and minimum meal frequency. Socioeconomic data were collected at 6, 12, 18 and 24 months. WAMI index (Water, sanitation, hygiene, Asset, Maternal education and Income index, ranging from 0 to 1) is a socioeconomic status index which includes access to improved water and sanitation, eight selected assets, maternal education, and household income was used as a representative of socioeconomic status of the households.[18] A higher WAMI index indicates a better socioeconomic status. To measure the household level crowding, a continuous variable named 'people per room' was generated by dividing the total number of people usually slept in the household by the number of rooms of that household used for sleeping. Blood samples for measuring serum zinc, ferritin and haemoglobin status were collected at 7, 15 and 24 months of age of a child. For haemoglobin, capillary blood was collected and measured with the Hemocue device (Hb 201, Ängelholm, Sweden). Plasma zinc and ferritin levels were measured using atomic absorption spectometry and chemiluminescence immunoassay, respectively. Anaemia is defined as a haemoglobin concentration <110 g/L.[19] Plasma ferritin and zinc concentration values were adjusted for the effect of subclinical inflammation using both C-reactive protein and α-1-acid glycoprotein.[20]

Trained field workers visited the participant's household twice a week and used a surveillance assessment form (SAF) to document the breastfeeding and illness status of the child. Household visits were generally made on every 3–4 days. During every such visits, field workers asked the caregiver whether the children experienced any of the symptoms listed in SAF on each day since the last visit.[21] To document the breastfeeding status, the staff asked the mother about the liquids the child consumed during the past 24 hours. If the response followed the WHO definition of exclusive breastfeeding (no other food or drink, not even water, except breast milk (including milk expressed, oral rehydration solution, drops and vitamins, minerals and medicines syrups)), then the child was considered as exclusively breast fed. Diarrhoea is defined as having three or more loose stools in a 24-hour period or at least one loose stool with blood reported by the mother.[22] A diarrhoeal episode is defined as being separated from another episode by at least two or more diarrhoea-free days.[22–24] Acute lower respiratory infection (ALRI) is identified based on the presence of cough and/or shortness of breath plus high respiratory rate.[24] The average of two respiratory rates was used, and was considered high based on the age of the child (<2 months of age: ≥60 breaths/minute; 2 to <12 months of age: ≥50 breaths/minute; ≥12 months of age: ≥40 breaths/minute).[25] ALRI episodes are separated by at least 15 ALRI-free days.[24] Fever is defined as an axillary temperature >37.5°C.[21]

## Statistics

We reported the maternal-level, child-level and household-level characteristics of the study participants using mean and SD for continuous variables and frequency as percentages for categorical variables. Student's t-test, $\chi^2$ test or the Fisher's exact test was used to compare the above-mentioned characteristics of the stunted children to their non-stunted counterparts at 24 months of age.

At stage 1, we analysed the longitudinal birth cohort data to identify the most predictive correlates of stunting. Serum micronutrient data were collected at 7, 15 and 24 months of age of the children. We carried forward 7 months data up to 14 months and 15 months data up to 23 months. Dietary intake data collection started at 9 months of age. Hence, to ensure the availability of maximum variables at every time point and temporality of the relation between outcome and explanatory variables, the longitudinal data analysis was done using the data collected from 9 to 24 months of age. Instead of exclusive breastfeeding (ebf) status (yes vs no), ebf days was used as it counts the specific number of days. The association between each explanatory variable and the binary outcome variable (stunting: no vs yes) was examined longitudinally using population-specific generalized estimating equations (GEE).[26] The GEE method is an extension of the quasi-likelihood approach that permits specification of a working correlation matrix for the within-subject correlation of repeated responses collected

from the same participants over time. As a result, more efficient and unbiased regression parameters are produced. Initially, bivariate analyses were performed to identify the unadjusted effect of each explanatory variable on the outcome variable. The explanatory variables showing statistically significant association (p<0.05) with the outcome variable in the bivariate analyses were selected for multivariable modelling. The variance inflation factor (VIF) was calculated to detect multicollinearity, and variables producing a VIF value >5 were excluded from the analysis. Quasi-likelihood information criterion (QIC) was used to select the correct covariance structure. The multivariable model with unstructured covariance matrix produced the smallest QIC value. Hence, we report the results of the multivariable model that was constructed using unstructured covariance matrix with robust variance estimates. We determined the strength of association by estimating the ORs and their 95% CIs. Statistical significance was determined at a p-value <0.05 and the variables showing statistically significant association with stunting in the multivariable GEE model were selected for the decomposition analysis.

At stage 2, threefold Blinder-Oaxaca decomposition analysis was done to measure the relative contribution of the identified variables to mean LAZ score difference between stunted and non-stunted children at 24 months of age. The threefold decomposition analysis decomposes the gap or difference of the group-specific means into three components: a gap due to differences in the level or magnitude of determinants or endowments, a gap due to the differences in the effects of the determinants or coefficients and a gap due to the interactions between endowments and coefficients.[27] During decomposition analysis, bootstrapped SEs were calculated based on 1000 replicates. R version 3.4.2 was used for data analysis and GEE and Oaxaca packages were employed.[28 29]

## Patient and public involvement statement

No patient or public were involved with the development of research question, designing the study, recruitment of participants, interpretation of the results and will be involved during disseminating the findings of the paper.

## RESULTS

At month 1, the cohort exhibited a 19% prevalence of stunting. The prevalence showed a similar trend until month 6 (18.4%). After that, this trend went up to 33.6%, 46.8% and 47.9% at 12, 18 and 24th months, respectively.

The sociodemographic characteristics of the participating households are described in table 1. Until 24 months, the study was able to follow a cohort of 211 children. At 24 months, the overall male to female ratio was 1.04, which was 1.3 in the stunted cohort. Only 10.9% of children were exclusively breast fed, the prevalence of which did not change over the stunting status. Most of the households were headed by a male family member. One-fifth of the mothers were illiterate and nearly half of

**Table 1**  Maternal-level, child-level, household-level characteristics and morbidity status of the cohort

| n=211 (at 24 months of age) | | All (n=211) | Stunted (n=101) | Not stunted (n=110) | |
|---|---|---|---|---|---|
| | | n (%) | | | P value |
| Maternal-, child- and household-level characteristics | | | | | |
| Sex | Male | 108 (51.2) | 57 (56.4) | 51 (46.4) | 0.1 |
| Exclusive breast feeding* | Yes | 23 (10.9) | 12 (11.9) | 11 (10.0) | 0.66 |
| Household head's sex | Male | 195 (92.4) | 95 (94.1) | 100 (90.9) | 0.83 |
| Mother's age at birth (years) | <18 years | 3 (1.4) | 0 (0.0) | 3 (2.7) | 0.2 |
| | 18–30 years | 178 (84.4) | 84 (83.2) | 94 (85.5) | |
| | >30 years | 30 (14.2) | 17 (16.8) | 13 (11.8) | |
| Mother's age at first pregnancy | <18 years | 84 (39.8) | 34 (33.7) | 50 (45.5) | 0.08 |
| Maternal education | No education | 41 (19.4) | 22 (21.8) | 19 (17.3) | 0.71 |
| | <5 years | 93 (44.1) | 43 (42.6) | 50 (45.5) | |
| | >5 years | 77 (36.5) | 36 (35.6) | 41 (37.3) | |
| Separate room for a kitchen | Yes | 17 (8.1) | 4 (4.0) | 13 (11.8) | 0.04 |
| Monthly family income | <US$100 | 94 (44.5) | 52 (51.5) | 42 (38.2) | 0.05 |
| Households have chicken or ducks | Yes | 13 (6.2) | 5 (5.0) | 8 (7.3) | 0.48 |
| Treat water to make it safe | Yes | 135 (64.0) | 56 (55.4) | 78 (70.9) | 0.02 |
| | | Mean (SD) | | | |
| Exclusive breastfeeding days | | 102.00 (57.7) | 105.00 (56.9) | 99.6 (58.5) | <0.00 |
| WAMI score† | | 0.53 (0.13) | 0.50 (0.13) | 0.55 (0.13) | 0.17 |
| People per room | | 3.57 (1.18) | 3.83 (1.32) | 3.33 (0.98) | <0.00 |
| Birth weight (kg) | | 2.82 (0.42) | 2.70 (0.39) | 2.92 (0.42) | <0.00 |
| Maternal height (cm) | | 149.03 (5.05) | 147.34 (4.85) | 150.54 (4.76) | <0.00 |
| Morbidity episodes/child-year | | | | | |
| Diarrhoea | | 3.63 | 4.03 | 3.43 | 0.08 |
| ALRI | | 0.47 | 0.54 | 0.46 | 0.98 |
| Fever | | 8.06 | 8.78 | 7.87 | 0.74 |
| Cough | | 9.69 | 9.64 | 9.83 | 0.02 |

*Excluding breast feeding is defined as no other food or drink, not even water, except breast milk (including milk expressed or from a wet nurse) for 6 months of life, but allows the infant to receive ORS, drops and syrups (vitamins, minerals and medicines).
†The WAMI score (ranging from 0 to 1) is a measure of household socioeconomic status including access to improved Water, sanitation and hygiene (WASH), Assets, Maternal education and Income.
ALRI, acute lower respiratory infection; ORS, oral rehydration solution.

them had an education of less than 5 years. Nearly half of the households had a monthly income of less than US$100, the percentage of which was higher in the stunted group than the non-stunted group. Mean birth weight of the stunted children was lower than their non-stunted counterparts. The mothers of the stunted children also had a lower height than those of the non-stunted children.

Morbidity data displayed in table 1 show that per year stunted children suffered from more than four diarrhoeal episodes which was higher than the overall annual rate of 3.63 episodes. Similar trends were seen for ALRI and fever episodes. However, the stunted and non-stunted differences in the diarrhoeal, ALRI and fever episodes are not statistically significant (p>0.05). Conversely,

the non-stunted group had a higher average of cough episodes (p=0.02) than their stunted counterparts.

Figure 1 represents the pattern of macronutrient intake from 9 to 24 months. Over the period, the trend of nutrient intake did not differ much between the stunted and non-stunted groups. For both the groups, carbohydrate was the main source of nutrition.

Figure 2 presents the comparative picture of prevalence of stunting among the children with and without MAD. Children without MADs were more stunted than children with MADs, and from 9 months onwards the difference in prevalence of stunting between these two groups increased with age. However, except at 15 and 23 months, the CIs of prevalence largely overlap. Hence, for

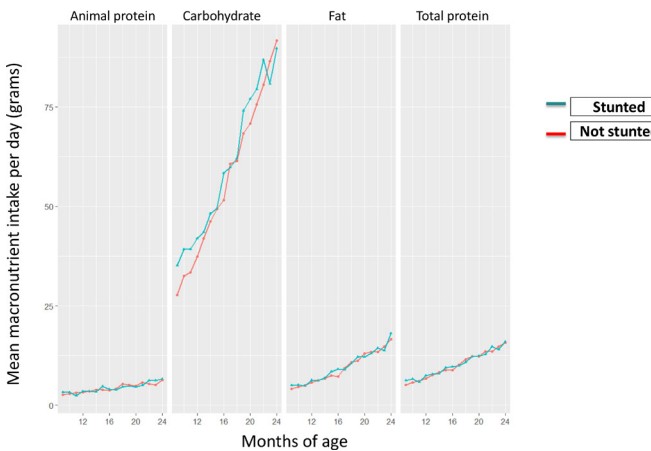

**Figure 1** Patterns of macronutrient intake from 9 to 24 months of age.

most of the months, the prevalence of stunting did not show statistically significant differences between the MAD and non-MAD groups.

Box plots reported in online supplementary figure 1 show the comparative picture of serum zinc, ferritin and haemoglobin levels of stunted and non-stunted children at 7, 15 and 24 months. The figure reveals that there were no notable differences in serum zinc, ferritin and haemoglobin levels between stunted and non-stunted children at 7, 15 and 24 months.

Table 2 presents the results of GEE modelling. Thorough literature review was done to identify the variables of interest, and bivariate GEE modelling was carried out longitudinally by taking into account all the variables, one at a time. However, table 2 shows only those variables found to have a statistically significant association (p<0.05) with stunting. From nutrient intake group, the amount of energy intake was selected because separate food groups (carbohydrate, total protein, animal protein and fat) showed a very high correlation with each other. The multivariable model showed significant (p<0.05)

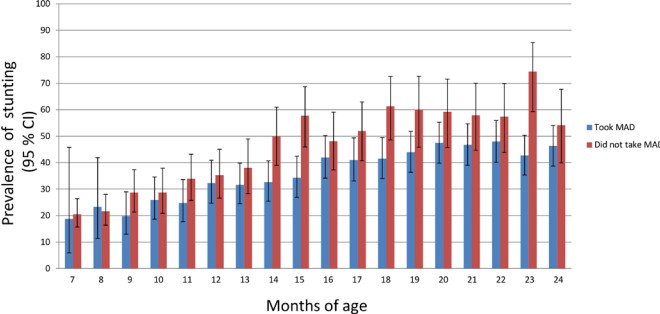

**Figure 2** Prevalence of stunting among children who did not take minimum acceptable diet (MAD) and those who took it (MAD: for breastfed children, MAD is defined as who had at least the minimum dietary diversity and the minimum meal frequency during the previous day and for non-breastfed children, it is defined as who received at least two milk feedings and had at least the minimum dietary diversity not including milk feeds and the minimum meal frequency during the previous day).

association of energy intake, gender, maternal height, having a separate room for kitchen, people per room and birth weight with stunting.

Table 3 and online supplementary figure 2 illustrate the results of Blinder-Oaxaca decomposition analysis. At month 24, the mean LAZ score of non-stunted group was −1.33 and in the stunted group it was −2.81, yielding a LAZ gap of 1.48. The stunted-non stunted LAZ gap had subsequently been decomposed into three components: a gap due to differences in the level or magnitude of determinants (endowments), a gap due to the differences in the effects of the determinants (coefficients) and a gap due to the interactions between endowments and coefficients. The results of the threefold decomposition suggest that, of the 1.48 difference, approximately 0.26 (95% CI: 0.41 to 0.10) was due to the group differences in endowments, 1.34 (95% CI: 1.50 to 1.19) to differences in coefficients and the remaining −0.12 (95% CI: 0.03 to -0.26) was for the interaction of the two. The 'endowment' part of the result reflects the average increase in stunted children's LAZ if they had the same level of characteristics as non-stunted children. This also indicates that the differences in the magnitude of explanatory variables account for about 17.6% of the LAZ gap among stunted and non-stunted children. The coefficient part shows that if the effect of the non-stunted children's characteristics was applied to the stunted children, the changes in the LAZ of the stunted cohort would be 1.34. The interaction component indicates that the simultaneous effect of endowment and coefficient was −0.12. The findings suggest that the gap in the means of LAZ is primarily due to the coefficient or effect of the determinants. The variable by variable decomposition of the LAZ gap identified maternal height (coefficient: −3.04; 95% CI: 0.35 to -6.44), birth weight (coefficient: −0.21; 95% CI: 0.88 to -1.30), people per room (coefficient: 0.31; 95% CI: 0.92 to -0.30) and energy intake (coefficient: −0.12; 95% CI: 0.22 to -0.46) as the top most factors responsible for the mean LAZ score difference between stunted and non-stunted at 24 months of age.

## DISCUSSION

At month 24, the cohort exhibited 47.9% prevalence of stunting. The figure is similar to Bangladesh Urban Health Survey which reported a 50% prevalence of stunting in under-five children living in slum areas of Dhaka, Bangladesh.[8]

Our analysis found maternal height to be inversely associated with stunting and exerting the highest relative contribution to LAZ score difference between stunted and non-stunted children at 24 months of age. Comparable findings were reported from studies conducted in similar setting. Results from 109 Demographic and Health Surveys conducted in 54 low/middle-income countries between 1991 and 2008 found maternal stature to be inversely associated with stunting during infancy and childhood.[30] It has also been published that a child,

**Table 2** Determinants of stunting among children: results of generalized estimating equation modelling (dependent variable: stunted no vs yes)

|  | Unadjusted OR (95% CI) | P value | Adjusted OR (95% CI) | P value |
|---|---|---|---|---|
| Energy intake* (kcal) | 1.001 (1.001 to 1.001) | 0.01 | 1.001 (1.001 to 1.002) | <0.00 |
| ebf days | 1.002 (1.001 to 1.003) | <0.00 | 1.002 (1.002 to 1.002) | 0.81 |
| Minimum acceptable diet | 0.74 (0.64 to 0.85) | <0.00 | 0.89 (0.72 to 1.09) | 0.25 |
| Cough episodes (per child-year) | 0.90 (0.82 to 0.98) | <0.00 | 0.97 (0.86 to 1.10) | 0.67 |
| Anaemia† | 1.45 (1.26 to 1.69) | <0.00 | 1.09 (0.90 to 1.32) | 0.40 |
| Gender | 0.70 (0.61 to 0.80) | <0.00 | 0.47 (0.38 to 0.58) | <0.00 |
| Maternal height (cm)* | 0.89 (0.88 to 0.91) | <0.00 | 0.87 (0.86 to 0.89) | <0.00 |
| Separate room for a kitchen | 0.37 (0.28 to 0.50) | <0.00 | 0.21 (0.12 to 0.38) | <0.00 |
| WAMI | 0.06 (0.04 to 0.11) | <0.00 | 0.39 (0.15 to 1.03) | 0.06 |
| People per room* | 1.34 (1.26 to 1.42) | <0.00 | 1.48 (1.35 to 1.62) | <0.00 |
| Birth weight (kg)* | 0.18 (0.15 to 0.22) | <0.00 | 0.11 (0.09 to 0.15) | <0.00 |
| Serum zinc (mmol/L) | 0.94 (0.91 to 0.98) | <0.00 | 0.99 (0.95 to 1.04) | 0.79 |

*Per one unit change.
†Anaemia is defined as a haemoglobin concentration <110 g/L.
ebf, exclusive breast feeding; WAMI, *W*ater,sanitation, hygiene, *A*sset, *M*aternal education and *I*ncome.

born to a shorter mother, had greater chances of being in a lower length-for-age category.[31] This could be explained by the biological plausibility that shorter mothers may have a poorer health and nutritional status and failed to deliver adequate nutrition to the fetus during pregnancy.[32] This finding signifies the intergenerational effect of malnutrition.

The negative association between birth weight and stunting identifies it as another important correlate of stunting. Moreover, the relative contribution of birth weight on mean LAZ score difference between stunted and non-stunted children at their 24 months of age indicates that its contribution did not decline with age. After birth, the mean LAZ score experiences a sharp fall, the magnitude of which can be affected by the presence of low birth weight—an indicator for underlying growth hindering factors.[33 34] A meta-analysis of 19 studies showed

that under-five children with low birth weight had three-fold higher odds of becoming stunted.[35]

Our findings suggest that children living in crowded households were more likely to be stunted. It also contributed to the mean LAZ sore difference between stunted and non-stunted children. This finding corresponds with the elucidations from a study done in Brazil which reported household people per room (OR: 2.40; 95% CI: 1.36 to 4.22) to be associated with a high prevalence of stunting.[36] Another study conducted in India showed that children living in crowded households (family with three or more persons per room) had a higher odds of being stunted than the children living in families with three members or less.[37] Having a higher number of family members may raise implications on household food insecurity or unavailability of complementary foods. Hence, the relationship warrants a

**Table 3** Relative contributions of the selected variables on mean length-for-age z score difference between stunted and non-stunted children at 24 months of age

|  | Endowment | | | Coefficient | | | Interaction | | |
|---|---|---|---|---|---|---|---|---|---|
|  |  | 95% CI | |  | 95% CI | |  | 95% CI | |
|  | Coefficient | Upper | Lower | Coefficient | Upper | Lower | Coefficient | Upper | Lower |
| Overall | 0.26 | 0.41 | 0.10 | 1.34 | 1.50 | 1.19 | −0.12 | 0.03 | −0.26 |
| Energy intake (kcal) | 0.00 | 0.03 | −0.03 | −0.12 | 0.22 | −0.46 | 0.00 | 0.03 | −0.03 |
| Gender | 0.01 | 0.02 | −0.01 | 0.02 | 0.11 | −0.07 | 0.00 | 0.01 | −0.02 |
| Maternal height (cm) | 0.10 | 0.18 | 0.02 | −3.04 | 0.35 | −6.44 | −0.07 | 0.02 | −0.15 |
| Separate kitchen | 0.00 | 0.03 | −0.03 | 0.01 | 0.03 | −0.02 | 0.02 | 0.06 | −0.03 |
| People per room | 0.06 | 0.12 | 0.00 | 0.31 | 0.92 | −0.30 | −0.04 | 0.04 | −0.12 |
| Birth weight (kg) | 0.08 | 0.17 | −0.01 | −0.21 | 0.88 | −1.30 | −0.02 | 0.07 | −0.11 |

further exploration keeping household food insecurity status as a covariate.

Though energy intake showed a marginal association with stunting in longitudinal data analysis, it did contribute to the LAZ gap at 24 months. This finding is supported by the multicountry analysis of MAL-ED birth cohort data which reported that lower per cent of energy from dietary protein was strongly associated with a decline in length-for-age.[31] Another study done in Zambia revealed that the stunted infants and toddlers had a tendency to take lower energy compared with the non-stunted children.[38]

We found that female children had lower odds of becoming stunted than their male counterparts. A meta-analysis of 16 demographic and health surveys done in sub-Saharan Africa reported boys to be more stunted than girls.[39] Similar results were found from East African herders where parental investments were biased for their female children.[40] On the contrary, studies also reported that in sub-Saharan areas, daughters faced a gender-biased dietary discrimination that was in favour of boys.[41] A cross-sectional analysis done in Bangladesh, Nepal and India documented that parental preference toward son might affect child undernutrition in this region.[42] However, there is a paucity of data that could elaborate this relation and the variable showed an insubstantial contribution in decomposition analysis.

In longitudinal analysis, having a separate room for kitchen was found to be negatively associated with stunting. Traditional cooking stoves inefficiently combust solid fuels and produce large volumes of indoor smoke.[43] Thus, using traditional cooking materials inside the living room acts as a potent source of indoor air pollution by producing life-threatening air pollutants.[44] This might have some biologically plausible relation with our finding. Multiple studies have reported air pollution to be associated with low birth weight.[43 45] However, no such report was found regarding stunting and the variable showed near zero contribution in decomposition analysis.

In addition to identifying the determinants of stunting, our study provided additional unique findings that the coefficients or effects of those determinants were primarily responsible for the stunted versus non-stunted LAZ gap at 24 months of age. That is the strength of this analysis. But, this study also has its own limitations to report. The study was conducted in a slum area. Hence, the findings of this study may not be applicable to the non-slum setting.

## CONCLUSIONS

The overall goal of the study was to measure the relative contributions of the most predictive correlates of stunting to mean LAZ score difference between stunted and non-stunted children at their 24 months of age. We aimed to generate concrete evidence in helping the policy-makers to direct their programme on specific areas of life cycle and to minimise the burden of linear growth faltering. In summary, we have documented that the affect of maternal height and birth weight on linear growth did not decline with age. Our findings indicated the intergenerational transmission of mother's own nutritional status to her offspring. The significance of this association at the policy level would thus ideally aim for the improvement in the nutritional status of a women during her adolescence and pregnancy, which would translate into positive inputs on the birth weight of her offspring and ultimately on the linear growth of the child.

**Author affiliations**
[1]Nutrition and Clinical Services Division, International Centre for Diarrhoeal Disease Research, Bangladesh (icddr,b), Dhaka, Bangladesh
[2]Maternal and Child Health Division (MCHD), International Centre for Diarrhoeal Disease Research, Bangladesh (icddr,b), Dhaka, Bangladesh

**Acknowledgements** The authors thank all the participants and their parents for sharing their time and providing consent, information and biological samples necessary for the successful completion of the study.

**Contributors** SD, SEA and TA conceived the study and procured funds. MM developed the data collections tools. MAA managed the data set and provided technical support. SD analysed the data, developed the tables/graphs and wrote the initial draft of the manuscript. MAA, MM, SEA and TA critically reviewed the manuscript and gave intellectual inputs. All authors contributed to the final version of the paper.

**Funding** This paper is made possible by the generous support of the American people through the US Agency for International Development (USAID) under cooperative agreement no: AID-388-A-17-00006. The MAL-ED birth cohort study was funded by University of Virginia with support from MAL-ED Network Investigators in the Foundation of National Institute of Health, Fogarty International Centre with overall support from the Bill & Melinda Gates Foundation. They also acknowledge the contribution of icddr,b's core donors including Government of the People's Republic of Bangladesh, Global Affairs Canada, Canada; Swedish International Development Cooperation Agency and the Department for International Development, UK Aid for their continuous support and commitment to the icddr,b's research efforts.

**Disclaimer** The views expressed herein are responsibility of the Research for Decision Makers (RDM) Activity and do not necessarily reflect the views of the US Government or USAID or icddr,b, the implementing agency.

**Competing interests** None declared.

**Patient consent for publication** Obtained.

**Ethics approval** Institutional Review Board, icddr,b (protocol no: PR-2008-020).

**Provenance and peer review** Not commissioned; externally peer reviewed.

**Data availability statement** The data sets generated and/or analysed during the current study is not made publicly available. However, data inquires or further suggestions for analyses can be made to the corresponding author.

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
