## [Reviewer comments · BMJ Open]

ARTICLE DETAILS

TITLE (PROVISIONAL)	Relative contributions of the correlates of stunting in explaining the mean length-for-age z-score difference between 24-month-old stunted and non-stunted children living in a slum of Dhaka, Bangladesh: Results from a decomposition analysis
AUTHORS	Das, Subhasish; Alam, Md Ashraf; Mahfuz, Mustafa; Arifeen, Shams E.; Ahmed, Tahmeed

VERSION 1 – REVIEW

REVIEWER	Daniel Corsi Children's Hospital of Eastern Ontario Research Institute and University of Ottawa
REVIEW RETURNED	27-Aug-2018

GENERAL COMMENTS	Overall this is a novel analysis and approach to consider the relative contribution of stunting in a cohort of young children in Bangladesh. I feel this manuscript warrants publication but the authors could make some clarifications and revisions which I feel will strengthen the manuscript and improve the communication of the findings. I have made some suggestions below according to the section of the paper. Title: consider shortening the title if possible to make it more concise Abstract: First sentence of objective is generic; consider removing. The objective is to determine the relative contribution of the most predictive correlates of stunting on explaining mean LAZ differences between stunted and non-stunted. Methods (abstract): Analysis uses 2 approaches. First determinants of stunting in this cohort were identified using the association of certain risk factors with stunting. Can you clarify if this was over time or at 24 months. If over time were you identifying changes in some of the exposure variables over time? Then a decomposition technique was used to determine the relative contribution of the most important covariates on explaining mean LAZ differences in stunted and non-stunted. Can you clarify how the variables were chosen to enter in the second stage either through some threshold of effect size and or statistical significance. There is a further discrepancy in that the outcome of the first part appears to be binary (stunted/non-stunted) while the decomposition uses a continuous outcome of LAZ. Results: Fair amount of jargon here if you can make this more approachable for those readers not familiar with the decomposition techniques. Your stated aim is to identify the determinants of stunting however all of the effects save people per room are presented as 'protective' factors for stunting. Consider using
---

	alternate reference categories so these can be presented in the same direction of effect. Main methods: Similar query to above can you elaborate on the change in outcome between the 2 analytical approaches. This could be set up in the manuscript introduction as well. It appears that only a P<0.05 threshold was used to consider inclusion of an explanatory variable in the adjusted model. Where both analyses run on the same dataset? Or was a restricted 'training' or 'testing' dataset created to identify the determinants of stunting in the first stage in a similar approach to risk prediction modelling? Any potential issues you are aware of in overfitting the decomposition model by not using the training/validation approach? This may be an issue for the discussion. Tables and figures: More definitions and footnotes are needed throughout. For example Table 1, please define WHO definition for exclusive breast feeding. Is mother's age current age? Or age when child is 24 months? Please clarify. Define IYCF Give duration of cough episodes -eg per year? Define anemia in table Add Units for maternal heigh, birth weight and serum zinc Figure 2: define MAD Supplemental Figure 1: Where any statistical or clinically important differences in these biomarkers identified? Can this be labelled in the figure? Supplemental Figure2: y-axis labels are not clear and need to be enlarged.
--	--

REVIEWER	Bradley A. Woodruff, MD MPH Groundwork LLC Switzerland
REVIEW RETURNED	14-Jan-2019

GENERAL COMMENTS	1) The authors want to prioritize those interventions which may alleviate or prevent stunting most efficiently. In order to accomplish this laudable goal, an analysis must determine which factors make the greatest contribution to stunting. However, the authors fail to explain why their methodology using Blinder-Oaxaca decomposition analysis is superior to other methods of achieving the same goal. For example, several methods exist to calculate relative risks from multi-variable modeling which can then be used to calculation population attributable risk. This method has been used in several studies of risk factors for various forms of malnutrition. Moreover, this reviewer does not understand how the contribution to stunting, as measured by LAZ, can be calculated separately in two groups themselves defined by an arbitrary cut-off of LAZ. What does such an analysis tell us that other types of analyses do not? 2) There are some small irregularities in language and writing. A thorough review by an English editor would be useful to further polish the text. a. Phrases which are abbreviated need not necessarily be themselves capitalized unless they are proper nouns. For example, "Generalized Estimating Equations (GEE) model" should be "generalized estimating equations (GEE) model", and "Minimum Acceptable Diet (MAD)" should be "minimum
--

	acceptable diet (MAD)". There are many other examples in the text of such mistaken capitalization. b. There are other example of awkward usage, such as lines 114-115 on page 6: "Plasma ferritin and zinc concentration values were adjustment for the effect of...". Another example is on line 129 of page 6: "The GEE models was constructed...". 3) How were serum zinc values adjusted for inflammation? The cited reference "UNICEF B. National Micronutrients Status Survey. 2013" is incomplete and unclear; I could not find anything on the web corresponding to this citation. Did the authors use one of the methods included in MacDonell Am J Clin Nutr 2018;107:932–940 or the regression modeling used in Karakochuk European Journal of Clinical Nutrition (2017) 71, 1467–1470? 4) Line 117 on page 6 says "Data on morbidity and infection status was collected twice a week." I assume the indicator is current diarrhea or LRI; however, the authors may wish to state this explicitly. Many readers will expect a cumulative incidence or prevalence over some recall period, such as the past 2 weeks because this is the more common indicator of morbidity in many cohort and cross-sectional data collections. 5) It may just be this reviewer's ignorance of generalized estimating equations, but how were the multiple measures of some independent variables, for example morbidity, SES status, food frequency, etc., related to a single measurement of the dependent outcome LAZ? And how was Blinder-Oaxaca analysis done with multiple measurements of each of several independent variables? This may need a more detailed explanation for less sophisticated readers like myself. 6) It would be nice to insert p values into Table 1. With only 101 and 110 children per group, it is difficult to tell without them which differences are really statistically significant and which are the result of sampling error only. Moreover, any statements in the manuscript that stunted children had more or less of a given characteristic than non-stunted children are not really valid without knowing if these differences in the study sample can be generalized to the population. 7) Many specific results are repeated in both Table 1 and the text of the Results section. This is not necessary. The manuscript should be reviewed to eliminate any such duplication. 8) The labels on the figures are incomplete. a. Figure 1 – The y axis label "Mean macronutrient intake per day" – is this in grams, kilocalories, or some other unit? All axis labels for continuous variables should include the unit of measurement. I assume the x axis label is months of age; however, "months" does not clearly say this. It could be months of follow-up or month of the year. b. Figure 2 – As above, the y axis label of "months" is incomplete. Moreover, what is the unit of the x axis and what does it show. There is no label at all. c. Figure 3 – What is the unit of zinc, ferritin, and hemoglobin on the y axes? What are the numbers 7, 15, and 24 at the top of each set of box plots? d. Figure 4 – The labels are completely illegible. The font is far too small, and the resolution is not good enough to allow magnification on my computer monitor. Increase the font size for all text in this figure at least 3-fold. 9) I do not think the results show in figure 2 are worth a figure. These results could be summarized with a simple sentence like "Children without minimally acceptable diets were more stunted than children with minimally acceptable diets, and the
--	---

	difference in stunting prevalence between these two groups increased with age.” Moreover, many specific results are repeated in this figure and in the text of the Result section. This is not necessary; the figure could be deleted and the text retained. 10) On line 178 of page 9 the text says that zinc concentrations are statistically significant different in stunted and non-stunted children. Figure 3 does not clearly show this. In fact, the differences between these two group at 7 and 15 months of age are minimal, and there appears to be no difference at all at 24 months of age. The authors my wish to explain this apparent contradiction in the text to avoid reader confusion. 11) Table 2 needs some revision: a. This reviewer does not understand the modeling. I assumed that the dependent variable in these models would be stunted or not or LAZ as a continuous variable. But the table includes separate p values for stunted and non-stunted children. How and why may need to be explained for less statistically sophisticated readers. b. I believe you are calculating odds ratios, not odds. The column headings should be adjusted accordingly. c. Why “Ebf days”? Many studies have shows that monther’s recall of when breastfeeding ended is unreliable. Moreover, the methods do not mention how often exclusive breastfeeding was measured; how can the precise number of days be counted if it is measured only occasionnally? d. What is “optimal IYCF”? It certainly does not consist only of minimum acceptable diet. There are 13 standard UNICEF/WHO indicators. I would interpret optimal IYCF as having the best values for all 13. e. Why do you now mention “cough episodes” now when the Results section describes in detail the definition of lower respiratory infection. f. Several of the comments above seem to indicate that you are trying to artificially make naturally categorical variables continuous variables. 12) Several statements in the discussion section are unclear: a. Several times the authors state that specific factors are associated with stunting; however, it is unclear whether you are referring to LAZ or to the classification stunted vs. not stunted. These are quite different indicators, so the distinction should be clear each time they are referred to. b. As described in the text, the study in Mexico cited on line 222 may not support this study’s findings regarding crowding and stunting. The text mentions that only family size was measured in the Mexico study, but this is not a measure of crowding. c. Line 235 says that female gender was negatively associated with stunting, but how? Female gender has only one value; therefore an association cannot be measured between an independent factor with only one value and the outcome stunting. d. Lines 235-238 cite studies showing more stunting in boys, but then state that there is a “female biased parenting investments”. Are these investments biased for or against girls?
--	--

REVIEWER	Catherine chwinger University of Bergen, Norway
REVIEW RETURNED	23-Jan-2019

GENERAL COMMENTS	General comment Overall, the authors address an important objective. Stunting is an immense public health problem with serious and long-lasting consequences. It is thus important to identify the most important risk factors. However, some points need to be addressed in this manuscript. The statistical methods need a more detailed description, especially the decomposition analysis and the interpretation thereof. The authors write that the findings of this study can be used to guide policy makers in Bangladesh. However, there is a need to discuss the generalizability of this study including 211 children living in a slum in Dhaka. Other issues, such as the loss to follow-up of ~20% and the possible effects on the results are missing in the discussion. The authors conclude that the findings affirm the need for “proper adolescent nutrition” to improve the stature. However, these conclusions cannot be drawn from the findings of this study. This need to be formulated more carefully as to my knowledge pure nutritional programs have not been shown to improve stature successfully.																
	Specific comments																
	   Lines Comment     35/36 Why is endowment reported in the results while it seems that the effect (coefficient) is more important in this sample?   38-40 The authors should be careful stating that there a strong contribution of birth weight while the 95% CI for the coefficient in the decomposition analysis is -0.21–0.89. I also question that one can conclude that “the results affirm the importance of nutritional programs”.   42-52 Bullet Point 2 is a repetition of the first, Bullet Point 4 and 5: Are these the most important limitations? Last point also unclearly formulated, what is the limitation with this?   56 To my knowledge, micronutrient deficiency is the most prevalent form of malnutrition. Please change sentence to “It is one of the most prevalent....” Also it is not known if it is the chronic malnutrition that is “responsible” for the high mortality, but stunting is an indicator/ is associated with higher mortality.   57-58 Please update numbers from the latest Global Nutrition Report (2018)   61-62 Unclear sentence. Why is the statement restricted to those children that have a normal anthropometric status at birth and to the period of up to 3 years of age?   65-68 The commonly used WHO framework for stunted growth and development could be mentioned for a more comprehensive overview over factors that could contribute to stunting   	Lines	Comment	35/36	Why is endowment reported in the results while it seems that the effect (coefficient) is more important in this sample?	38-40	The authors should be careful stating that there a strong contribution of birth weight while the 95% CI for the coefficient in the decomposition analysis is -0.21–0.89. I also question that one can conclude that “the results affirm the importance of nutritional programs”.	42-52	Bullet Point 2 is a repetition of the first, Bullet Point 4 and 5: Are these the most important limitations? Last point also unclearly formulated, what is the limitation with this?	56	To my knowledge, micronutrient deficiency is the most prevalent form of malnutrition. Please change sentence to “It is one of the most prevalent....” Also it is not known if it is the chronic malnutrition that is “responsible” for the high mortality, but stunting is an indicator/ is associated with higher mortality.	57-58	Please update numbers from the latest Global Nutrition Report (2018)	61-62	Unclear sentence. Why is the statement restricted to those children that have a normal anthropometric status at birth and to the period of up to 3 years of age?	65-68	The commonly used WHO framework for stunted growth and development could be mentioned for a more comprehensive overview over factors that could contribute to stunting
	Lines	Comment															
	35/36	Why is endowment reported in the results while it seems that the effect (coefficient) is more important in this sample?															
	38-40	The authors should be careful stating that there a strong contribution of birth weight while the 95% CI for the coefficient in the decomposition analysis is -0.21–0.89. I also question that one can conclude that “the results affirm the importance of nutritional programs”.															
	42-52	Bullet Point 2 is a repetition of the first, Bullet Point 4 and 5: Are these the most important limitations? Last point also unclearly formulated, what is the limitation with this?															
	56	To my knowledge, micronutrient deficiency is the most prevalent form of malnutrition. Please change sentence to “It is one of the most prevalent....” Also it is not known if it is the chronic malnutrition that is “responsible” for the high mortality, but stunting is an indicator/ is associated with higher mortality.															
	57-58	Please update numbers from the latest Global Nutrition Report (2018)															
	61-62	Unclear sentence. Why is the statement restricted to those children that have a normal anthropometric status at birth and to the period of up to 3 years of age?															
65-68	The commonly used WHO framework for stunted growth and development could be mentioned for a more comprehensive overview over factors that could contribute to stunting																

	(Stewart et al 2013, Maternal and Child Nutrition)
70-72	It is unclear what the cost-benefit ratio is and the numbers reported cannot be found in the literature cited.
72-74	How can the results of this specific study guide policy makers in Bangladesh?
126	Were all continuous variables normally distributed?
129	Why is the unstructured covariance matrix used and could this have implications for the results?
131-132	It is unclear what "low frequency" means
133	Please try to avoid fill-words such as "so" (also "next" line 201)
137-138	It is unclear how the selected variables were entered into the multivariable model (all at the same time)? Please give more details here.
151-167	The information in Table 1 does not need to be repeated in the text again. Suggestion: Maybe the authors could pick out the most interesting characteristics. If it is stated that for example a proportion is higher in the stunted group without any formal statistical testing, the numbers in each group should be given in the text. The numbers for ALRI in the text (line 167) differ from the numbers in the table!
171/172	"taking" a diet sounds incorrect, please use a different phrasing
174	The prevalence was not higher at 8 months of age (please correct: "but every month the non-MAD group had a higher prevalence of stunting than the MAD group").
177-179	The author write that serum zinc had a statistical significant difference between the group. Which test was used? This is not described in the methods. This is not expected looking at Figure 2.
180	An a priori conceptual framework is mentioned, but this is not described in the manuscript.
187	If days of EBF was used instead of EBF status, this measure could also be described in Table 1 and the method section.
190-203	Please add the 95% CIs. How did you calculate the 16% in line 199. If this is 0.26/1.46 this should be 17.6% To my knowledge, the decomposition method has not been extensively used in this research area and I expect that it is not well known by many readers. I would appreciate a better description of the meaning of the results presented in Table 3 in the discussion section.
218	What were the findings of this study. Please expand on what was confirmed.
238-239	The literature review on gender differences in malnutrition in Asia is incomplete. Please expand. As example Raj et al 2014: "Gendered Effects of Siblings on Child

		Malnutrition in South Asia: Cross-sectional Analysis of Demographic and Health Surveys from Bangladesh, India, and Nepal” can be mentioned.
	N/A-Discussion section	Please include a discussion on the generalizability and the potential effects of 20% loss to follow-up in this cohort. Please discuss the use and the results of the decomposition analysis.
	246-253	The recommendation to provide adequate nutrition is not based on the findings of this analysis.
	Table 2	Please include the units in the table. Why is the higher number mentioned first in the 95%? This is difficult to read.
	Figure 1	What is the unit of measurement on the y-axis?
	Figure 2	What is the unit on the x-axis? X-axis title is missing. Could error bars be added? This is subjective, but I would prefer to switch the axis so that age in months would be on the x-axis.
	Suppl. Figure 1	Please consider excluding extreme outliers to make the scale of the y-axis more appropriate. Please include units of measurements (y-axis). Please indicate what 7, 15 and 24 means.
	Suppl. Figure 2	This graph is unreadable! Please revise!

VERSION 1 – AUTHOR RESPONSE

Reviewer: 1

Reviewer Name: Daniel Corsi

Institution and Country: Children's Hospital of Eastern Ontario Research Institute and University of Ottawa

Please state any competing interests or state 'None declared': None declared

Please leave your comments for the authors below

Overall this is a novel analysis and approach to consider the relative contribution of stunting in a cohort of young children in Bangladesh. I feel this manuscript warrants publication but the authors could make some clarifications and revisions which I feel will strengthen the manuscript and improve the communication of the findings. I have made some suggestions below according to the section of the paper.

Title: consider shortening the title if possible to make it more concise

Response: Thank you very much for reviewing the manuscript. We have renamed the manuscript as- “Relative contributions of the correlates of stunting in explaining the mean length-for-age z-score difference between 24-month-old stunted and non-stunted children living in a slum of Dhaka, Bangladesh: Results from a decomposition analysis”.

Abstract: First sentence of objective is generic; consider removing. The objective is to determine the relative contribution of the most predictive correlates of stunting on explaining mean LAZ differences between stunted and non-stunted.

Response: We have edited the section accordingly.

Methods (abstract): Analysis uses 2 approaches. First determinants of stunting in this cohort were identified using the association of certain risk factors with stunting. Can you clarify if this was over time or at 24 months. If over time were you identifying changes in some of the exposure variables over time? Then a decomposition technique was used to determine the relative contribution of the most important covariates on explaining mean LAZ differences in stunted and non-stunted. Can you clarify how the variables were chosen to enter in the second stage either through some threshold of effect size and or statistical significance? There is a further discrepancy in that the outcome of the first part appears to be binary (stunted/non-stunted) while the decomposition uses a continuous outcome of LAZ.

Response: Thank you very much for raising this issue. We identified the determinants of stunting using the data collected longitudinally from 9 to 24 months of age of the study participants. We identified the changes of the exposure variables over time. We used 'p value <0.05' as the cutoff for statistical significance to choose the variables for second stage. For selecting the candidate variables for second stage analysis, we could use any conceptual framework. But, as we had the longitudinal data available from the same children, we rather analyzed the data using generalized estimating equations (GEE) model, identified the most predictive correlates of stunting (where the comparison group was non-stunted children), and then we applied the decomposition technique to determine the relative contribution of those covariates on explaining mean LAZ differences in stunted and non-stunted children at 24 months their age using decomposition technique. Hope this explains the cause of discrepancy in the outcome variables (binary and continuous) for the two analyses.

Results: Fair amount of jargon here if you can make this more approachable for those readers not familiar with the decomposition techniques. Your stated aim is to identify the determinants of stunting however all of the effects save people per room are presented as 'protective' factors for stunting. Consider using alternate reference categories so these can be presented in the same direction of effect.

Response: We concur with your concern regarding the presence of jargons. We have rewritten the section accordingly. For identifying the determinants of stunting, we have used "stunted: no" as the reference category. For the continuous variable 'people per room', we have found that the OR is 1.48. That means- with one unit increase in people per room, the likelihood of getting stunted increases by 1.48 times.

Main methods: Similar query to above can you elaborate on the change in outcome between the 2 analytical approaches. This could be set up in the manuscript introduction as well. It appears that only a $P < 0.05$ threshold was used to consider inclusion of an explanatory variable in the adjusted model. Where both analyses run on the same dataset? Or was a restricted 'training' or 'testing' dataset created to identify the determinants of stunting in the first stage in a similar approach to risk prediction modeling? Any potential issues you are aware of in over fitting the decomposition model by not using the training/validation approach? This may be an issue for the discussion.

Response: We used the longitudinal data collected from 9 to 24 months for identifying the determinants of stunting. Then, decomposition analysis was done using the 24 months' data- a cross-sectional set of the main panel data. Instead of doing so, we could select the candidate variables from any conceptual framework, but, as we had longitudinal data available from the same cohort, we rather analyzed the data using generalized estimating equations (GEE) model, identified the most predictive correlates of stunting (where the comparison group was non-stunted children), and then we applied the decomposition technique. Our aim was to identify the correlates of stunting that best fit the existing cohort. We did not create any training or testing data set for the first stage analysis. We could not identify any potential issues that can cause over fitting of the decomposition model. But, we calculated bootstrapped standard errors based on 1,000 replicates to ensure robustness of the result, Tables and figures:

More definitions and footnotes are needed throughout. For example Table 1, please define WHO definition for exclusive breast feeding. Is mother's age current age? Or age when child is 24 months? Please clarify. Define IYCF

Give duration of cough episodes -eg per year? Define anemia in table. Add Units for maternal height, birth weight and serum zinc.

Figure 2: define MAD

Supplemental Figure 1: Where any statistical or clinically important differences in these biomarkers identified? Can this be labelled in the figure?

Supplemental Figure2: y-axis labels are not clear and need to be enlarged.

Response: Thank you very much. We have edited the tables, figures and the corresponding sections of the text accordingly. The bivariate GEE modeling revealed that, over the months, serum zinc and hemoglobin levels had statistically significant (p value <0.05) association with stunting. We have edited the corresponding section of the text accordingly.

Reviewer: 2

Reviewer Name: Bradley A. Woodruff, MD MPH

Institution and Country: Groundwork LLC - Switzerland

Please state any competing interests or state 'None declared': None

Please leave your comments for the authors below

1. The authors want to prioritize those interventions which may alleviate or prevent stunting most efficiently. In order to accomplish this laudable goal, an analysis must determine which factors make the greatest contribution to stunting. However, the authors fail to explain why their methodology using Blinder-Oaxaca decomposition analysis is superior to other methods of achieving the same goal. For example, several methods exist to calculate relative risks from multi-variable modeling which can then be used to calculation population attributable risk. This method has been used in several studies of risk factors for various forms of malnutrition. Moreover, this reviewer does not understand how the contribution to stunting, as measured by LAZ, can be calculated separately in two groups themselves defined by an arbitrary cut-off of LAZ. What does such an analysis tell us that other types of analyses do not?

Response: Thank you very much for reviewing our paper. We used the 'Threefold Blinder-Oaxaca' decomposition analysis to examine the relative contribution of the identified variables on mean length-for-age Z score difference between stunted and non-stunted children at 24 months of age. Multiple studies have measured the role of different risk factors of stunting using odds ratio, relative risk and other risk estimates. All of these estimates concentrate only on the effects of the determinants or coefficients values. Hence, other than the mean, detailed decomposition of other parameters cannot be done. So, we used 'Threefold Blinder-Oaxaca' decomposition analysis that decomposes the role of the determinants into three components: a gap due to differences in the level or magnitude of determinants or endowments, a gap due to the differences in the effects of the determinants or coefficients, and a gap due to the interactions between endowments and coefficients.

2. There are some small irregularities in language and writing. A thorough review by an English editor would be useful to further polish the text.

a. Phrases which are abbreviated need not necessarily be themselves capitalized unless they are proper nouns. For example, "Generalized Estimating Equations (GEE) model" should be "generalized estimating equations (GEE) model", and "Minimum Acceptable Diet (MAD)" should be "minimum acceptable diet (MAD)". There are many other examples in the text of such mistaken capitalization.

Response: Thank you very much. We have edited the texts accordingly.

b. There are other example of awkward usage, such as lines 114-115 on page 6: “Plasma ferritin and zinc concentration values were adjustment for the effect of...”. Another example is on line 129 of page 6: “The GEE models was constructed...”.

Response: Thank you very much. We have edited the texts accordingly.

3. How were serum zinc values adjusted for inflammation? The cited reference “UNICEF B. National Micronutrients Status Survey. 2013” is incomplete and unclear; I could not find anything on the web corresponding to this citation. Did the authors use one of the methods included in MacDonell *Am J Clin Nutr* 2018;107:932–940 or the regression modeling used in Karakochuk *European Journal of Clinical Nutrition* (2017) 71, 1467–1470?

Response: Serum ferritin and zinc values were adjusted for infection by estimating biomarkers of infection: CRP and AGP. We followed the method described by Frederick et al., 2011. We are extremely sorry for providing the incorrect reference. We have edited the section accordingly. The details of this approach can be found here- Grant FK, Suchdev PS, Flores-Ayala R, Cole CR, Ramakrishnan U, Ruth LJ, Martorell R. Correcting for Inflammation Changes Estimates of Iron Deficiency among Rural Kenyan Preschool Children-. *The Journal of nutrition*. 2011 Dec 7;142(1):105-11.

4. Line 117 on page 6 says “Data on morbidity and infection status was collected twice a week.” I assume the indicator is current diarrhea or LRI; however, the authors may wish to state this explicitly. Many readers will expect a cumulative incidence or prevalence over some recall period, such as the past 2 weeks because this is the more common indicator of morbidity in many cohort and cross-sectional data collections.

Response: Thank you very much for raising this issue. We have edited the section accordingly. Daily illness surveillance data were collected twice a week in the participant's household by trained MAL-ED field staff using the ‘Surveillance Assessment Form’. Field staff members communicated verbally with the mothers or other caregivers in their local languages and asked the mother or other caregiver whether the children experienced any of the symptoms or took any of the treatments listed in the form on each day since the last study visit. Household visits were generally made every 3–4 days. The details of this approach can be found here- Richard SA, Barrett LJ, Guerrant RL, Checkley W, Miller MA. Disease surveillance methods used in the 8-site MAL-ED cohort study. *Clinical Infectious Diseases*. 2014 Nov 1;59(suppl_4):S220-4.

5. It may just be this reviewer's ignorance of generalized estimating equations, but how were the multiple measures of some independent variables, for example morbidity, SES status, food frequency, etc., related to a single measurement of the dependent outcome LAZ? And how was Blinder-Oaxaca analysis done with multiple measurements of each of several independent variables? This may need a more detailed explanation for less sophisticated readers like myself.

Response: The generalized estimating equations (GEE) method is an extension of the quasi-likelihood approach that permits specification of a working correlation matrix for the within-subject correlation of repeated responses collected from the same participant over time. As a result, more efficient and unbiased regression parameters are produced. We used GEE models to measure the association of multiple measures of the independent variables to the multiple measures of the binary dependent outcome ‘stunting (no vs. yes)’. Field workers measured the recumbent length of the children every month until 24 months of age. On each occasions, the LAZ score of each child was determined using the WHO 2006 Child Growth Standards and a LAZ < -2 was classified as stunted. Blinder-Oaxaca analysis was done on mean length-for-age Z score difference between stunted and non-stunted children at 24 months of age only. Our objective was to determine the relative

contribution of the most predictive correlates of stunting on explaining mean LAZ differences between stunted and non-stunted at 24 months of age.

6. It would be nice to insert p values into Table 1. With only 101 and 110 children per group, it is difficult to tell without them which differences are really statistically significant and which are the result of sampling error only. Moreover, any statements in the manuscript that stunted children had more or less of a given characteristic than non-stunted children are not really valid without knowing if these differences in the study sample can be generalized to the population.

Response: Thank you very much. We have inserted the p values into the table.

7. Many specific results are repeated in both Table 1 and the text of the Results section. This is not necessary. The manuscript should be reviewed to eliminate any such duplication.

Response: Thank you very much. We have edited the section according to your valuable suggestion.

8. The labels on the figures are incomplete.
 - a. Figure 1 – The y axis label “Mean macronutrient intake per day” – is this in grams, kilocalories, or some other unit? All axis labels for continuous variables should include the unit of measurement. I assume the x axis label is months of age; however, “months” does not clearly say this. It could be months of follow-up or month of the year.

Response: Thank you very much. We have edited the figure accordingly.

- b. Figure 2 – As above, the y axis label of “months” is incomplete. Moreover, what is the unit of the x axis and what does it show. There is no label at all.

Response: Thank you very much. We have edited the section accordingly and, following the suggestion of Catherine Schwinger (reviewer 3), we have switched the axis and put the age in months on the x-axis.

- c. Figure 3 – What is the unit of zinc, ferritin, and hemoglobin on the y axes? What are the numbers 7, 15, and 24 at the top of each set of box plots?

Response: Thank you very much. We have edited the figure accordingly.

- d. Figure 4 – The labels are completely illegible. The font is far too small, and the resolution is not good enough to allow magnification on my computer monitor. Increase the font size for all text in this figure at least 3-fold. –

Response: We have edited the figure to make it readable. Thank you.

- e. I do not think the results show in figure 2 is worth a figure. These results could be summarized with a simple sentence like “Children without minimally acceptable diets were more stunted than children with minimally acceptable diets, and the difference in stunting prevalence between these two groups increased with age.” Moreover, many specific results are repeated in this figure and in the text of the Result section. This is not necessary; the figure could be deleted and the text retained.

Response: We have edited the figure and the corresponding section in the text accordingly. Following the suggestion of Catherine Schwinger (reviewer 3), we have added 95% CI bars to make the figure more informative.

9. On line 178 of page 9 the text says that zinc concentrations are statistically significant different in stunted and non-stunted children. Figure 3 does not clearly show this. In fact, the differences between these two group at 7 and 15 months of age are minimal, and there appears to be no difference at all at 24 months of age. The authors my wish to explain this apparent contradiction in the text to avoid reader confusion.

Response: The bivariate GEE modeling revealed that, over the months, serum zinc and hemoglobin levels had statistically significant (p value <0.05) association with stunting. We have edited the corresponding section of the text accordingly to explain this contradiction.

10. Table 2 needs some revision:

- a. This reviewer does not understand the modeling. I assumed that the dependent variable in these models would be stunted or not or LAZ as a continuous variable. But the table includes separate p values for stunted and non-stunted children. How and why may need to be explained for less statistically sophisticated readers.

Response: Here, the dependent variable is stunted (no vs. yes). The table reports p values generated from bivariate (column 3) and multivariate (column 5) GEE analysis.

- b. I believe you are calculating odds ratios, not odds. The column headings should be adjusted accordingly.

Response: Column headings are edited accordingly. Thank you.

- c. Why “Ebf days”? Many studies have shown that mother’s recall of when breastfeeding ended is unreliable. Moreover, the methods do not mention how often exclusive breastfeeding was measured; how can the precise number of days be counted if it is measured only occasionally?

Response: Instead of exclusive breast feeding (ebf) status (yes vs. no), we used ‘exclusive breast feeding days (ebf days)’ for the final model. We have seen that there are some children who were exclusively breast fed for more than 170 days. So if we use ‘exclusive breast feeding (ebf) status (yes vs. no)’, then they fall into the ‘non-exclusively breast fed’ criteria. Instead if we use the continuous variable ‘ebf days’ in our analysis, then it covers the entire spectrum.

Trained field staff collected data on child’s breast feeding status twice a week in the participant’s household using the Surveillance Assessment Form. The staff asked the mother about the liquids the child consumed during the past 24 hours. If the response went in line with the WHO definition of exclusive breast feeding (no other food or drink, not even water, except breast milk (including milk expressed, ORS, drops and vitamins, minerals and medicines syrups)), then the child was considered as exclusively breast fed. A copy of the questionnaire can be found below-

11. Several statements in the discussion section are unclear:

a. Several times the authors state that specific factors are associated with stunting; however, it is unclear whether you are referring to LAZ or to the classification stunted vs. not stunted. These are quite different indicators, so the distinction should be clear each time they are referred to.

Response: We have revised the section accordingly. Thank you very much.

b. As described in the text, the study in Mexico cited on line 222 may not support this study's findings regarding crowding and stunting. The text mentions that only family size was measured in the Mexico study, but this is not a measure of crowding.

Response: We have revised the reference accordingly.

c. Line 235 says that female gender was negatively associated with stunting, but how? Female gender has only one value; therefore an association cannot be measured between an independent factor with only one value and the outcome stunting.

Response: Thank you very much for identifying this issue. We have revised the section accordingly. We categorized the gender into two groups- male and female and 'male' gender is the reference value for the reported analysis. We have found that female children had lesser chances (aOR: 0.47) of becoming stunted than their male counterparts.

d. Lines 235-238 cite studies showing more stunting in boys, but then state that there is a "female biased parenting investments". Are these investments biased for or against girls?

Response: We pointed to the investments biased for girls. We have edited the section to make it more comprehensive.

Reviewer: 3

Reviewer Name: Catherine Schwinger

Institution and Country: University of Bergen, Norway

Please state any competing interests or state 'None declared': None declared

Please leave your comments for the authors below

Please find my comments in the document attached

Comments to the authors

General comment

Overall, the authors address an important objective. Stunting is an immense public health problem with serious and long-lasting consequences. It is thus important to identify the most important risk factors.

However, some points need to be addressed in this manuscript. The statistical methods need a more detailed description, especially the decomposition analysis and the interpretation thereof. The authors write that the findings of this study can be used to guide policy makers in Bangladesh. However, there is a need to discuss the generalizability of this study including 211 children living in a slum in Dhaka. Other issues, such as the loss to follow-up of ~20% and the possible effects on the results are missing in the discussion. The authors conclude that the findings affirm the need for "proper adolescent nutrition" to improve the stature. However, these conclusions cannot be drawn from the findings of this study. This need to be formulated more carefully as to my knowledge pure nutritional programs have not been shown to improve stature successfully.

Response: Thank you very much for reviewing the manuscript. We have edited the manuscript to reflect all your suggestions. Please find the point-to-point responses below-
Specific comments

Lines	Comment
35/36	Why is endowment reported in the results while it seems that the effect (coefficient) is more important in this sample? Response: Thank you very much for your suggestion. We have edited the section accordingly.
38-40	The authors should be careful stating that there a strong contribution of birthweight while the 95% CI for the coefficient in the decomposition analysis is -0.21–0.89. I also question that one can conclude that “the results affirm the importance of nutritional programs”. Response: Thank you very much for your suggestion. We have edited the section accordingly.
42-52	Bullet point 2 is a repetition of the first, Bullet Point 4 and 5: Are these the most important limitations? Last point also unclearly formulated, what is the limitation with this? Response: We have edited the section accordingly.
56	To my knowledge, micronutrient deficiency is the most prevalent form of malnutrition. Please change sentence to “It is one of the most prevalent....” Also it is not known if it is the chronic malnutrition that is “responsible” for the high mortality, but stunting is an indicator/ is associated with higher mortality. Response: We have edited the section accordingly.
57-58	Please update numbers from the latest Global Nutrition Report (2018) Response: We have edited the section accordingly. Thank you.
61-62	Unclear sentence. Why is the statement restricted to those children that have a normal anthropometric status at birth and to the period of up to 3 years of age? Response: We have edited the statement to make it more general. Thank you.
65-68	The commonly used WHO framework for stunted growth and development could be mentioned for a more comprehensive overview over factors that could contribute to stunting (Stewart et al 2013, Maternal and Child Nutrition) Response: We have edited the section accordingly. Thank you very much for your valuable suggestion.
70-72	It is unclear what the cost-benefit ratio is and the numbers reported cannot be found in the literature cited. Response: We have edited the citation accordingly. We are extremely sorry for citing a different literature.
72-74	How can the results of this specific study guide policy makers in Bangladesh? Response: Since Bangladesh is a resource constrained country, identifying the top contributors of stunting would help the policy-makers to direct concerted stunting prevention programs in a cost effective way. We have edited the section accordingly to reflect this thought.
126	Were all continuous variables normally distributed? Response: Thank you very much for pointing out this issue. We have removed the words “normally distributed”.

129	Why is the unstructured covariance matrix used and could this have implications for the results? Response: Quasi-information criterion (QIC) was used to select the correct covariance structure. The multivariable model with unstructured covariance matrix produced the smallest QIC value. Hence, we report the results of the multivariable model that was constructed using unstructured covariance matrix with robust variance estimates. We have edited the statistical analysis section accordingly.
131-132	It is unclear what “low frequency” means Response: We have removed the words. Thank you.
133	Please try to avoid fill-words such as “so” (also “next” line 201) Response: We have edited the section accordingly. Thank you.
137-138	It is unclear how the selected variables were entered into the multivariable model (all at the same time)? Please give more details here. Response: The association between each explanatory variable and outcome variable (stunting: no vs. yes) was examined longitudinally using Generalized Estimation Equations (GEE) model. The explanatory variables showing statistically significant association ($p < 0.05$) with the outcome variable in the bivariate analysis were selected for multivariable modeling. We have edited the section accordingly. We have edited the statistical analysis section accordingly.
151-167	The information in Table 1 does not need to be repeated in the text again. Suggestion: Maybe the authors could pick out the most interesting Characteristics. If it is stated that for example a proportion is higher in the stunted group without any formal statistical testing, the numbers in each group should be given in the text. The numbers for ALRI in the text (line 167) differ from the numbers in the table! Response: We have edited the section accordingly. And, we now report p-value of the statistical testing in table 1.
171/172	“taking” a diet sounds incorrect, please use a different phrasing Response: We have edited the section accordingly. Thank you.
174	The prevalence was not higher at 8 months of age (please correct: “but every month the non-MAD group had a higher prevalence of stunting than the MAD group”). Response: We have removed the sentence.
177-179	The authors write that serum zinc had a statistical significant difference between the group. Which test was used? This is not described in the methods. This is not expected looking at Figure 2. Response: We are extremely sorry for creating the confusion. The bivariate GEE modeling revealed that serum zinc and hemoglobin levels had statistically significant (p value < 0.05) association with stunting. We have edited the section accordingly.
180	An a priori conceptual framework is mentioned, but this is not described in the manuscript. Response: A thorough literature review was done to identify the variables of interest. The use of the term “a priori conceptual framework” is redundant. We have edited the section carefully. We are extremely sorry for creating the confusion.
187	If days of EBF were used instead of EBF status, this measure could also be described in Table 1 and the method section. Response: Edited accordingly. Thank you very much.

190-203	Please add the 95% CIs. How did you calculate the 16% in line 199. If this is 0.26/1.46 this should be 17.6% To my knowledge, the decomposition method has not been extensively used in this research area and I expect that it is not well known by many readers. I would appreciate a better description of the meaning of the results presented in Table 3 in the discussion section. Response: Sorry for the typo. We have corrected it accordingly. We have rewritten the result section to make it more meaningful to the readers.
218	What were the findings of this study. Please expand on what was confirmed. Response: We have edited the section accordingly. Thank you very much.
238-239	The literature review on gender differences in malnutrition in Asia is incomplete. Please expand. As example Raj et al 2014: "Gendered Effects of Siblings on Child Malnutrition in South Asia: Cross-sectional Analysis of Demographic and Health Surveys from Bangladesh, India, and Nepal" can be mentioned. Response: We have edited the section accordingly. Thank you very much.
N/A- Discussion section	Please include a discussion on the generalizability and the potential effects of 20% loss to follow-up in this cohort. Please discuss the use and the results of the decomposition analysis. Response: We have edited the section accordingly. Thank you very much.
246-253	The recommendation to provide adequate nutrition is not based on the findings of this analysis. Response: We have removed the sentence. Thank you very much.
Table 2	Please include the units in the table. Response: Edited accordingly. Thank you very much.
	Why is the higher number mentioned first in the 95%? This is difficult to read. Response: Edited accordingly. Thank you very much.
Figure 1	What is the unit of measurement on the y-axis? Response: The unit of the measurement is grams. We have mentioned it now. Thank you very much.
Figure 2	What is the unit on the x-axis? X-axis title is missing. Could error bars be added? This is subjective, but I would prefer to switch the axis so that age in months would be on the x-axis. Response: Thank you very much for sharing your insights. We have edited the figure accordingly. Following your suggestion, we have added 95% CI bars to make the figure more informative. Thank you.
Suppl. Figure 1	Please consider excluding extreme outliers to make the scale of the y-axis more appropriate Please include units of measurements (y-axis). Please indicate what 7, 15 and 24 means. Response: We have edited the figure accordingly. Thank you very much.
Suppl. Figure 2	This graph is unreadable! Please revise! Response: We have edited the figure to make it readable. Thank you very much.

VERSION 2 – REVIEW

REVIEWER	Catherine Schwinger University of Bergen, Norway
REVIEW RETURNED	04-Mar-2019

GENERAL COMMENTS	The manuscript has improved and I have only some minor specific comments. Line 17: Please be consistent with the study acronym: MAL-ED or Mal-ED throughout the manuscript Line 104: Can you extend the explanation of WAMI, e.g. what does an increase in WAMI mean? Line 134: Please correct “frequency and percentages” to “frequency as percentages” Lines 187-189: The authors state that the group of stunted children had more episodes of diarrhea, ALRI, fever and cough. However, the difference are NOT statistically significant and for cough it is the non-stunted group who had a higher average of cough episodes ($p = 0.02$). Please clarify this in the text. Lines 193-196: The interpretation of Figure 2 needs refinement. Although it appears that children with MAD had a higher prevalence of stunting compared to without MAD, the CI largely overlap (except at 15 and 23 months). Lines 197-200: Could the authors expand on the explanation? It seems that there is no difference in serum zinc, ferritin and hemoglobin between stunted and not stunted children according to Supplementary Figure 1. It is then surprising that the GEE shows that they do predict stunting. Line 246: This sentence is unclear, birthweight cannot be described as “growth hindering factor” but might be a good indicator for underlying growth hindering factors. Line 263: Please change “We found that female children had lesser chances of becoming...” to “We found that female children had lower odds of becoming...” Line 283: Could the authors explain in more detail how the 20% loss to follow-up could lead to bias? Line 295: The authors conclude that sufficient energy intake should be provided, but the variable energy intake was chosen because separate food groups showed a high collinearity (lines 204-206). Therefore, energy intake could just be a confounder and it is not clear if it is the provision of energy or the provision of a single nutrient group that could lead to an improvement in linear growth. Please revise the sentence!
--

VERSION 2 – AUTHOR RESPONSE

Reviewer: 3

Reviewer Name: Catherine Schwinger

Institution and Country: University of Bergen, Norway

Please state any competing interests or state 'None declared': None declared

Please leave your comments for the authors below:

The manuscript has improved and I have only some minor specific comments.

Dear Catherine Schwinger,

Thank you very much for taking your time to review the manuscript. Please find below our brief responses (in italics) for your kind review and consideration:

1. Line 17: Please be consistent with the study acronym: MAL-ED or Mal-ED throughout the manuscript.

Response: Thank you very much for your observation. We have edited the line accordingly.

2. Line 104: Can you extend the explanation of WAMI, e.g. what does an increase in WAMI mean?

Response: WAMI or Water, sanitation, hygiene, Assets, Maternal education and Income is an indicator of socio-economic status of the study participants. The index ranges from 0 to 1 and its higher value indicates better socio-economic status. We have edited the section accordingly to represent this.

3. Line 134: Please correct “frequency and percentages” to “frequency as percentages”.

Response: Corrected accordingly. Thank you very much.

4. Lines 187-189: The authors state that the group of stunted children had more episodes of diarrhea, ALRI, fever and cough. However, the difference are NOT statistically significant and for cough it is the non-stunted group who had a higher average of cough episodes ($p = 0.02$). Please clarify this in the text.

Response: We have edited the text accordingly. Thank you very much.

5. Lines 193-196: The interpretation of Figure 2 needs refinement. Although it appears that children with MAD had a higher prevalence of stunting compared to without MAD, the CI largely overlap (except at 15 and 23 months).

Response: We completely agree to your comment. We think that, for most the months, the prevalence of stunting are not significantly different between the MAD and non-MAD groups. We have edited the section accordingly to reflect this thought.

6. Lines 197-200: Could the authors expand on the explanation? It seems that there is no difference in serum zinc, ferritin and hemoglobin between stunted and not stunted children according to Supplementary Figure 1. It is then surprising that the GEE shows that they do predict stunting.

Response: Thank you very much for digging out the ambiguity. We have removed the line and focused only on explaining the figure specific information.

7. Line 246: This sentence is unclear, birthweight cannot be described as “growth hindering factor” but might be a good indicator for underlying growth hindering factors.

Response: We have edited the section accordingly. Thank you very much.

8. Line 263: Please change “We found that female children had lesser chances of becoming...” to “We found that female children had lower odds of becoming...”.

Response: We have edited the section accordingly. Thank you very much.

9. Line 283: Could the authors explain in more detail how the 20% loss to follow-up could lead to bias?

Response: Thank you very much for pointing out the issue. We think the line adds much confusion. Hence, we considered removing the line.

10. Line 295: The authors conclude that sufficient energy intake should be provided, but the variable energy intake was chosen because separate food groups showed a high collinearity (lines 204-206). Therefore, energy intake could just be a confounder and it is not clear if it is the provision of energy or the provision of a single nutrient group that could lead to an improvement in linear growth. Please revise the sentence!

Response: We have removed the line to make the entire section clearer to the readers. Thank you very much for your keen observation and comment.